# How the Softmax Activation Hinders the Detection of Adversarial and Out-of-Distribution Examples in Neural Networks

## Abstract

Despite having excellent performances for a wide variety of tasks, modern neural networks are unable to provide a prediction with a reliable confidence estimate which would allow to detect misclassifications. This limitation is at the heart of what is known as an adversarial example, where the network provides a wrong prediction associated with a strong confidence to a slightly modified image. Moreover, this overconfidence issue has also been observed for out-of-distribution data. We show through several experiments that the softmax activation, usually placed as the last layer of modern neural networks, is partly responsible for this behaviour. We give qualitative insights about its impact on the MNIST dataset, showing that relevant information present in the logits is lost once the softmax function is applied. The same observation is made through quantitative analysis, as we show that two out-of-distribution and adversarial example detectors obtain competitive results when using logit values as inputs, but provide considerably lower performances if they use softmax probabilities instead: from 98.0% average AUROC to 56.8% in some settings. These results provide evidence that the softmax activation hinders the detection of adversarial and out-of-distribution examples, as it masks a significant part of the relevant information present in the logits.

## 1 Introduction

Thanks to their excellent performances, Neural Networks (NN) are now used to tackle important problems such as medical diagnosis (Shen et al., 2017) or pedestrian detection for autonomous cars (Tian et al., 2015). However, regarding classification, one of the issues that prevent widespread adoption of such solutions is their overconfidence in their predictions (Amodei et al., 2016). It has been observed that they can provide predictions with high confidence values for errors (Guo et al., 2017), out-of-distribution (OOD) examples such as tailored noise (Nguyen et al., 2015) and adversarial examples (Szegedy et al., 2014).

The softmax layer has been identified as one potential culprit for these overconfident predictions (Hendrycks & Gimpel, 2017; Neumann et al., 2018). The softmax function is defined as follows:

$$\sigma(z_i) = \frac{e^{z_i}}{\sum_{j=1}^{C} e^{z_j}} \quad \text{for } i = 1, ..., C \quad \text{and} \quad \boldsymbol{z} \in \mathbb{R}^C$$

Hendrycks & Gimpel (2017) state that overconfident prediction are due to the use of the fast-growing exponential function in the softmax computation. It is thus designed to output high maximum values, even for small differences in the logits $\boldsymbol{z}$. Solutions have been proposed for this specific issue, such as *temperature scaling*, which divides the logits $\boldsymbol{z}$ by a constant $T > 1$ in order to obtain "softer" probabilities. Several works showed that this method significantly improves the calibration of the probabilities (Guo et al., 2017; Neumann et al., 2018).

However, another property of the softmax, which is less discussed, is that the outputs are normalized in order to obtain a probability distribution. These probabilities are indeed easier to interpret than logits, as humans can process probabilities intuitively (Cosmides & Tooby, 1996). However, the information about the logits absolute values is lost due to this normalization. To the best of our

knowledge, only the very recent work of Ozbulak et al. (2019) adress this issue. In their work, they state that some adversarial examples, referred to as "over-optimized" adversarial examples, have very high logit values, but their magnitude is masked by the softmax function. They thus use this difference in logit distribution to detect this type of adversarial examples. Our work is aligned with theirs and extends their findings, as we show that logits can be used not only to detect over-optimized adversarial examples, but also "regular" adversarial examples and OOD data. We also propose new experiments to quantify the impact of the loss of the information contained in the logits on detection performances.

Our main contrutions are :

- We provide some intuition about the difference between logit and softmax distributions through experiments on MNIST. We show that there is a significant difference in logit distributions between regular MNIST images and OOD and adversarial images, but this difference is not present to the same extent after the softmax function is applied.

- We propose 2 logit-based solutions inspired by state-of-the-art methods, each being suited for a different scenario: a NN trained on logit activations when the adversarial attacks and OOD datasets are known beforehand, and a kernel density estimation of the logit distribution when they are not. We achieve competitive performances for the detection of 4 kinds of adversarial attacks and 4 OOD datasets. These results remains true for 2 datasets, MNIST and CIFAR10, and 2 NN architectures, a custom, relatively shallow one and a Wide Residual Network (Zagoruyko & Komodakis, 2016). It is noteworthy that few detectors are evaluated on both adversarial and OOD examples, as most work focus on only one problem.

- Finally, we show that using the softmax probabilities in place of the logit activations results in significantly worse performances for both detectors. This highlights that relevant information is lost once the softmax function is applied, and raises questions about its role in the lack of robustness observed in NN.

The ouline of the paper is as follows. In Section 2, we present a brief summary of state-of-the-art methods for 3 related problems: adversarial example detection, out-of-distribution example detection and uncertainty estimation. Section 3 provides some qualitative insights about the difference in distribution between logit and softmax values on MNIST. In Section 4, we show quantitatively the detrimental impact of the softmax on OOD and adversarial detectors performances. We summarize our findings and propose future directions in Section 5.

## 2 RELATED WORK

### 2.1 ADVERSARIAL EXAMPLE DETECTION

In response to the growing number of adversarial attacks that have appeared in the literature recently (Chen & Jordan, 2019; Goodfellow et al., 2015; Szegedy et al., 2014; Moosavi-Dezfooli et al., 2016), detecting adversarial examples has also become a hot topic (Feinman et al., 2017; Lee et al., 2018; Metzen et al., 2017; Ozbulak et al., 2019; Xu et al., 2018). Detection methods can be split into 3 categories: detecting changes in sample statistics (Feinman et al., 2017; Lee et al., 2018; Ozbulak et al., 2019), training a detector (Metzen et al., 2017) and looking for prediction inconsistency (Xu et al., 2018).

Feinman et al. (2017) propose to use a combination of density estimates on the last hidden layer and bayesian uncertainty estimates (computed using dropout during inference (Gal & Ghahramani, 2016)) to determine whether an exemple has been adversially perturbed or not. Metzen et al. (2017) train an adversarial example detector network using intermediate feature representations as inputs. Ozbulak et al. (2019) show that over-optimized adversarial examples have much higher logit values than legitimate samples, and thus propose to use a threshold on these values for detection. Xu et al. (2018) check the consistency of a NN predictions between a given input and a squeezed version of it to differentiate adversarial and regular images. Lee et al. (2018) use the Mahalanobis distance on NN learned representations to detect both adversarial and OOD examples. It is noteworthy that their detector is, to the best of our knowledge, one of the only other detectors that have been evaluated on these 2 tasks.

## 2.2 OUT-OF-DISTRIBUTION EXAMPLE DETECTION

Another related challenge that has recently received attention from the research community is the detection of OOD examples (DeVries & Taylor, 2018; Hendrycks & Gimpel, 2017; Lee et al., 2018; Liang et al., 2017; Vyas et al., 2018). Indeed, it has been shown that a NN trained on ImageNet can provide predictions with almost 100% confidence for images that are just noise to the human eye (Nguyen et al., 2015). Hendrycks & Gimpel (2017) present a framework to evaluate misclassified and OOD example detectors and show that softmax probabilities can detect both of them to some extent, but that there is room for improvement. DeVries & Taylor (2018) propose to learn confidence estimates that discriminates between in and out-of-distribution examples by allowing NNs to ask for "hints" during training when the prediction is uncertain. Liang et al. (2017) introduce ODIN, a method that separates in and out-of-distribution examples by preprocessing the input using adversarial perturbation and then thresholding the softmax scores computed after temperature scaling. Vyas et al. (2018) train an ensemble of classifiers with a margin-based term added to the loss to detect OOD examples.

## 2.3 UNCERTAINTY ESTIMATION

A closely related and more general field of research is uncertainty estimation. It can be used in different settings such as semi-supervised learning (Triguero et al., 2015), weighting an ensemble of classifiers (Mandelbaum & Weinshall, 2017), or misclassification detection (Jiang et al., 2018; Mandelbaum & Weinshall, 2017; Papernot & McDaniel, 2018). Regarding this last task, it is noteworthy that all 3 works are strongly inspired by the KNN algorithm. Jiang et al. (2018) introduce the Trust Score, a specific 1-NN ratio applied on a filtered version of the training set in order to recognize misclassified examples. In the same spirit, Mandelbaum & Weinshall (2017) propose a confidence score based on the distance in the hidden representation space between an examples and its K nearest neighbors in the training set. Papernot & McDaniel (2018) compute a confidence score based on the homogeneity of the set of nearest neighbors computed in the representation space of each layer.

## 3 DIFFERENCE BETWEEN LOGIT AND SOFTMAX DISTRIBUTIONS

In this Section, we provide intuition about the difference between logit and softmax values distributions through experiments on the MNIST dataset (LeCun et al., 1998). Our results show that there are significant differences in logit activations between MNIST, OOD and adversarial examples, but that these differences are less or not present at all once the softmax function has been applied.

### 3.1 SETUP

**NN model:** For these experiments, we train a simple custom NN using the Keras framework (Chollet et al., 2015). The architecture is described in Table 1. The training was done for 30 epochs. We used the RMSprop optimizer, a batch size of 64 and data augmentation (i.e. slight rotations, zooms and shifts).

**OOD and adversarial examples:** In order to study how logit and softmax values are distributed, we use or generate the following OOD and adversarial datasets:

- For out-of-distribution examples, we study 4 different kinds of data. We use random normal and uniform noise images, as well as images from CIFAR-10 (Krizhevsky et al., 2014) and Fashion MNIST (Xiao et al., 2017) datasets. This allows to study 3 different kinds of OOD examples: noise images, natural images drawn from a significantly different distribution (CIFAR-10), and images drawn from a relatively close distribution (Fashion-MNIST).

- For adversarial examples, we use the CleverHans framework (Papernot et al., 2018). We generate 4 adversarial datasets using different methods: FGSM (Goodfellow et al., 2015), BIM (Chen & Jordan, 2019), DeepFool (Moosavi-Dezfooli et al., 2016) and the MaxConfidence attack (Goodfellow et al., 2019). FGSM and BIM are two well known attacks, while DeepFool and MaxConfidence are complementary: DeepFool aims at finding the minimal perturbation which changes the prediction, while MaxConfidence aims at finding

Table 1: Architecture of the custom NN.

| LAYER TYPE | PATCH SIZE | STRIDE | DEPTH | PADDING | ACTIVATION | OUTPUT SIZE |
|---|---|---|---|---|---|---|
| CONVOLUTION | 3x3 | 1 | 32 | NONE | RELU | 26x26x32 |
| CONVOLUTION | 3x3 | 1 | 32 | NONE | RELU | 24x24x32 |
| MAX POOLING | 2x2 | 2 | | | | 12x12x32 |
| DROPOUT (20%) | | | | | | 12x12x32 |
| CONVOLUTION | 3x3 | 1 | 64 | SAME | RELU | 12x12x64 |
| CONVOLUTION | 3x3 | 1 | 64 | SAME | RELU | 12x12x64 |
| MAX POOLING | 2x2 | 2 | | | | 6x6x64 |
| DROPOUT (25%) | | | | | | 6x6x64 |
| CONVOLUTION | 3x3 | 1 | 128 | SAME | RELU | 6x6x128 |
| DROPOUT (25%) | | | | | | 6x6x128 |
| FLATTEN | | | | | | 4608 |
| DENSE | | | | | RELU | 128 |
| BATCH NORMALIZATION | | | | | | 128 |
| DROPOUT (25%) | | | | | | 128 |
| DENSE | | | | | SOFTMAX | 10 |

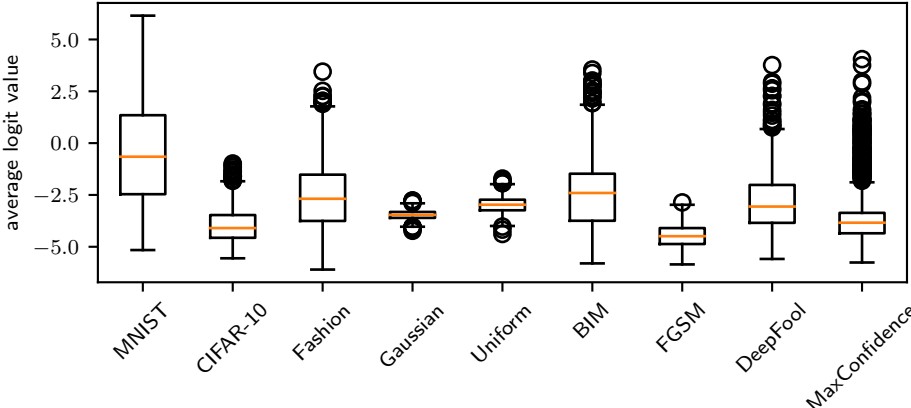

Figure 1: Distribution of the average logit values for 1000 examples of each dataset.

the perturbation which results in the highest confidence for a wrong class. We only keep the images that are misclassified by the NN. All the images have been generated using the $l_\infty$ norm with $\epsilon = 0.3$.

## 3.2 ANALYSIS OF THE DISTRIBUTION OF LOGIT AND SOFTMAX VALUES

Figure 1 displays the distribution of the average logit values for MNIST, OOD and adversarial images. We can clearly see that logits values are higher for MNIST images than for any other dataset. These differences are statistically significant according to the Student's t-test ($p < 0.0001$ between MNIST and any other dataset). Whereas for softmax values, the average is always $1/C$ with $C$ the number of classes, as the sum of the probabilities over all the classes is equal to 1. Thus, we can see that because of the normalization of the softmax function, we lose the information about the average logit value, which appears to be discriminating.

Figure 2 presents a scatterplot of the logit and softmax minimum and maximum values for the MNIST, Gaussian, CIFAR-10, FGSM and MaxConfidence datasets. The Figure displays only these 5 datasets for the sake of visibility. Exhaustive results are presented in the Appendix. Qualitatively, we can clearly distinguish cluster of points representing each dataset. Hence, it appears that these 2 simple statistics for logits are enough to separate fairly well legitimate examples from OOD and

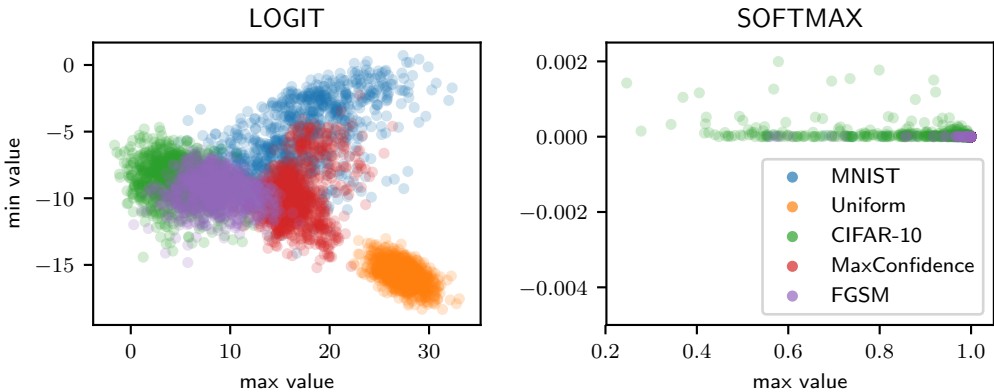

Figure 2: Scatterplot of the logit and softmax minimum and maximum values. Best viewed in color.

adversarial ones. We can see that legitimate images tend to have higher minimum and/or maximum logit values, which correlates well with the distributions showed in Figure 1. However, we can see that the discriminating information is once again lost when we apply the softmax function, as the exponential function combined with the normalization is pushing the highest value towards 1 and the lowest one towards 0.

These observations give some insights about the detrimental effect of the softmax function regarding OOD and adversarial example detection. The valuable information about the average of the logit value is lost due to the normalization, while the exponential function pushes minimum output values towards 0 and maximum ones toward 1, making it difficult to determine if an example is legitimate or not.

## 4 OOD AND ADVERSARIAL DETECTION

In this Section, we propose and evaluate 2 simple detection solutions inspired by state-of-the-art methods: a NN detector trained on both legitimate and illegitimate examples, and a kernel density estimator (KDE) fitted only on legitimate inputs. We first show that these 2 methods perform at a competitive level when using logit activations. Then, we show that using softmax probabilities instead of logit activation results in a significant drop in performance.

### 4.1 FIRST METHOD: NN-BASED DETECTOR

Similarly to the solution proposed by Metzen et al. (2017), but with a simpler architecture, we train a 3 layers regression NN with the considered representation (either logit of softmax values) as input. It aims at predicting the confidence value associated to a given prediction, i.e. whether it is a correct (value of 1) or incorrect (value of 0). The NN is composed of 3 dense layers with 128 neurons and RELU activations. The first 2 layers are followed by dropout layers with a dropout rate of 20%, and the second dropout layer is also followed by a Batch Normalization layer. We train the network for 60 epochs using RMSProp to optimize the mean squared error loss. It is noteworthy that only one NN is trained on all OOD and adversarial datasets. One could train one NN per kind of illegitimate examples, but we chose to train a generic one instead.

### 4.2 SECOND METHOD: KDE-BASED DETECTOR

Based on the solution proposed by Feinman et al. (2017), we fit a KDE on the considered representation of legitimate inputs. To do so, we use the implementation provided by the scipy python package (Jones et al., 2001). The main parameter of a KDE is the bandwidth, as it manages whether the resulting estimated distribution is going to be "spiky" or "smooth". The scipy implementation uses the Scott's rule (Scott, 2015) to compute its value.

### 4.3 EXPERIMENTAL SETUP

**Experiments:** We evaluate both methods on 2 datasets, MNIST and CIFAR-10, and on 2 NN architectures. The first one is the custom network presented in Section 3, the second one is a Wide Residual Network (Zagoruyko & Komodakis, 2016), with depth 28 and width 8, whose implementation is provided by Keras contrib. We evaluate each combination of NN architecture $\times$ dataset, resulting in 4 experimental settings. We do so to ensure that our results can be generalized to several conditions. Regarding classification performances, the custom network achieves 99.6% accuracy on MNIST and 80.2% on CIFAR-10, while the Wide Residual Network achieves 99.7% on MNIST and 95.2% on CIFAR-10.

**Evaluation metrics:** We use the evaluation framework introduced in Hendrycks & Gimpel (2017): we compute the Area Under the Receiver Operating Characteristic curve (AUROC), the Areas Under the Precision-Recall curve (AUPR) for both the correct (AUPR In) and incorrect (AUPR Out) classes. In addition, we also compute the False Positive Rate at 95% True Positive Rate (FPR) as in Liang et al. (2017). For clarity reasons, we only display the average of these metrics over OOD and adversarial datasets.

**Data:** For the NN-based detector, the training set is composed of 5000 examples belonging to the in-distribution dataset (i.e. MNIST or CIFAR-10), and of 1000 examples of all the remaining datasets presented in Section 3 (MNIST is used as an OOD dataset when CIFAR-10 is the considered in-distribution dataset). These numbers were chosen in order to have a fairly balanced dataset. From this data, we use 10% as a validation set. For the KDE detector, we only use the 5000 legitimate examples to fit the distribution. In both cases, the testing set is composed of 2000 images of each dataset.

### 4.4 COMPARISON WITH STATE OF THE ART METHODS

In this first set of experiments, we compare the 2 methods we presented, using logit activations as inputs, with 2 detectors proposed in the literature. The first method is the baseline proposed by Hendrycks & Gimpel (2017), which consists in thresholding on the max softmax probabilities to determine whether an example is legitimate or not. We refer to this method as the Softmax Baseline. The second method is the Trust Score introduced in Jiang et al. (2018). To compare these detectors, we mostly focus on the AUROC metric, as it is the most comprehensive metric among the 4 considered.

Results are presented in Table 2. The NN detector outperforms the other methods for all settings, most of the time by a large margin. It achieves results close to perfection on the MNIST dataset, with average AUROC over 99% for both architectures. It is especially true for OOD examples, for which it obtains an average AUROC of 99.9%. Regarding the CIFAR-10 dataset, it obtains slighty lower results for the detection of adversarial examples, with AUROC averages close to 95%. However, this detector is the only one which has the advantage of knowing beforehand the kind of illegitimate examples it is going to be evaluated on. It is thus expected that it would provide the best results. Nevertheless, it shows that logits contain the required information to detect adversarial and OOD examples with high accuracy.

Regarding the other 3 methods, we can see that the Trust Score and the KDE solution achieve overall comparable performances. When comparing the average AUROC on both adversarial and OOD examples, they achieve similar performances on both datasets with the Wide Residual Network architecture: 98.0% on MNIST and 82.6% on CIFAR-10 for the Trust Score, 98.0% on MNIST and 82.7% on CIFAR-10 for the KDE. Their performances differ slightly with the custom architecture, as the Trust Score achieves better results on MNIST (91.6% compared to 89.7%), but lower ones on CIFAR-10 (71.1% compared to 79.7%). When looking at the detailed results, it appears that the KDE detector struggles with detecting OOD examples on CIFAR-10, as it is significantly outperformed by the Softmax Baseline for both architectures. However, this observation does not hold for the MNIST dataset, as the KDE obtain the second best results for OOD example detection on both architectures. It also vastly outperforms the Trust Score on adversarial example detection on CIFAR-10.

These results show that both the NN and the KDE detectors achieve results which are competitive with state of the art methods. This observation remains true for different architectures and datasets, showing that these methods can be relevant OOD and adversarial example detectors *when provided with the relevant inputs*.

Table 2: OOD and adversarial example detection results on MNIST and CIFAR-10, for both the custom NN and the Wide Residual Network (referred to as WR-28-8). The best result for each metric/experience, in italic, is always achieved by the NN-based detector. However, as this detector is the only one which has the advantage of knowing beforehand the OOD and adversarial datasets, we also highlight the second best results in bold.

| EXPERIMENT | DATASETS | FPR (95% TPR) | AUROC | AUPR IN | AUPR OUT |
|---|---|---|---|---|---|
| | | SOFTMAX BASELINE / TRUST SCORE / KDE / NN | | | |
| CUSTOM NETWORK MNIST | ADVERSARIAL | 54.2/**47.3**/59.0/*6.4* | 72.8/**90.4**/83.0/*98.4* | 80.2/**92.6**/85.3/*98.5* | 73.0/**86.3**/78.6/*98.0* |
| | OOD | 58.1/43.5/**24.2**/*0.2* | 59.5/92.8/**96.4**/*99.9* | 73.7/95.0/**97.0**/*99.9* | 62.1/87.2/**95.2**/*99.9* |
| | ALL | 56.2/45.4/**41.4**/*3.3* | 66.2/**91.6**/89.7/*99.2* | 76.9/**93.8**/91.2/*99.2* | 67.6/86.7/**86.9**/*99.0* |
| WR-28-8 MNIST | ADVERSARIAL | 63.5/23.2/**20.3**/*3.3* | 65.8/96.8/**96.9**/*99.3* | 78.6/**97.6**/97.6/*99.5* | 64.8/95.5/**95.8**/*98.8* |
| | OOD | 60.6/**3.0**/3.4/*0.1* | 65.5/**99.1**/99.1/*99.9* | 79.4/**99.3**/99.3/*99.9* | 60.0/**98.9**/98.9/*99.9* |
| | ALL | 62.0/13.1/**11.9**/*1.7* | 65.6/**98.0**/98.0/*99.6* | 79.0/**98.5**/98.4/*99.7* | 62.4/97.2/**97.4**/*99.4* |
| CUSTOM NETWORK CIFAR-10 | ADVERSARIAL | 90.9/90.1/**36.7**/*35.7* | 35.2/63.7/**87.7**/*94.1* | 52.9/71.2/**87.9**/*95.0* | 45.1/56.8/**85.7**/*91.2* |
| | OOD | 87.5/83.5/**69.1**/*5.4* | 78.0/**79.6**/71.6/*98.6* | **83.5**/83.5/77.8/*98.6* | 69.4/**72.9**/71.7/*98.7* |
| | ALL | 89.2/86.8/**52.9**/*20.6* | 56.6/71.1/**79.7**/*96.4* | 68.2/77.3/**82.9**/*96.8* | 57.3/64.8/**78.7**/*95.0* |
| WR-28-8 CIFAR-10 | ADVERSARIAL | 67.2/76.6/**33.5**/*30.4* | 53.9/71.2/**91.2**/*95.7* | 69.8/76.9/**91.9**/*96.6* | 57.8/66.6/**89.3**/*93.4* |
| | OOD | 51.4/**33.4**/49.1/*1.2* | 91.7/**94.0**/74.3/*99.6* | 93.8/**94.9**/76.2/*99.7* | 87.5/**92.8**/73.0/*99.6* |
| | ALL | 59.3/55.0/**41.3**/*15.8* | 72.8/82.6/**82.7**/*97.7* | 81.8/**85.9**/84.0/*98.1* | 72.7/79.7/**81.2**/*96.5* |

## 4.5 THE DETRIMENTAL EFFECT OF THE SOFTMAX FUNCTION

In this second set of experiments, we compare the performances of the KDE and NN detectors depending on whether the logit or softmax values are used as inputs. Using both detectors is interesting as they operate in different conditions. On one hand, the KDE detector operates in an unsupervised setting (i.e. the OOD and adversarial datasets are not known in advance) and is using the "raw" values to fit the distribution. On the other hand, the NN detector represents the best-case scenario, as it operates in a supervised setting and can extract the most relevant features out of the inputs for the detection task.

Results of these experiments are summarized in Table 3. Using the logits as inputs instead of softmax probabilities provides *better results in the vast majority of settings*, with the only exception of detecting OOD examples on CIFAR-10 with the KDE solution, for which softmax values obtain slightly better performances (average AUROC of 76.0% compared to 74.3% for logits). This is consistent with the results presented in Table 2, as the Softmax Baseline outperformed the KDE detector for OOD example detection on CIFAR-10. However, the logit version of the NN detector outperform the softmax version in the same setting (average AUROC of 99.6% compared to 96.8%). One possible explanation is that "raw" logits might not be refined enough to separate legitimate and OOD examples, but extracting features from them fixes this issue to a greater extent than for softmax probabilities.

Regarding all the other settings, using the logits as input leads to significantly better results. It is especially true for the KDE detector, for which we observe considerable relative improvements in average AUROC: +59% on MNIST and +32% on CIFAR-10 for the custom architecture, +72% and +19% for the Wide Residual Network. Regarding the NN detector, we observe slightly lower relative improvements: +9% on MNIST and +14% on CIFAR-10 for the custom NN, +12% and +10% for the Wide Residual Network. However, the FPR at 95% TPR provides intuitive insights about what these differences in AUROC means. For instance, regarding the custom NN on MNIST for which we observed a relative improvement of 9% by using logits instead of softmax for the NN detector, the FPR decreases from 55.1% to 3.3%. Hence, the seperation of legitimate and illegitimate examples seems to be considerably better for an average AUROC of 99.2% than for an average of 90.2%.

Looking at the detailed results, it appears that it is more detrimental to use softmax probabilities for detecting adversarial examples than for detecting OOD examples. If we consider the NN detector, the improvement obtained by using the logits is indeed higher for adversarial than for OOD examples. A

Table 3: Comparison of detection performances between using the softmax and logits values as input for both the KDE and NN detectors. For each detector, the best result of each settings is in bold.

| EXPERIMENT | DATASETS | FPR (95% TPR) | | AUROC | | AUPR IN | | AUPR OUT | |
|---|---|---|---|---|---|---|---|---|---|
| | | KDE SOFTMAX/KDE LOGITS | NN SOFTMAX/NN LOGITS | KDE | NN | KDE | NN | KDE | NN |
| CUSTOM NETWORK MNIST | ADVERSARIAL | 67.8/**59.0** | 58.9/**6.4** | 67.3/**83.0** | 86.6/**98.4** | 69.8/**85.3** | 88.9/**98.5** | 66.0/**78.6** | 81.8/**98.0** |
| | OOD | 72.6/**24.2** | 51.4/**0.2** | 45.1/**96.4** | 95.2/**99.9** | 56.5/**97.0** | 97.3/**99.9** | 53.0/**95.2** | 79.6/**99.9** |
| | ALL | 70.2/**41.6** | 55.1/**3.3** | 56.2/**89.7** | 90.9/**99.2** | 63.1/**91.2** | 93.1/**99.2** | 59.5/**86.9** | 80.7/**99.0** |
| WR-28-8 MNIST | ADVERSARIAL | 66.2/**20.3** | 60.0/**3.3** | 65.3/**96.9** | 80.4/**99.3** | 64.9/**97.6** | 83.5/**99.5** | 67.4/**95.8** | 75.9/**98.8** |
| | OOD | 74.9/**3.4** | 3.6/**0.1** | 48.3/**99.1** | 97.1/**99.9** | 59.6/**99.3** | 97.8/**99.9** | 58.9/**98.9** | 84.5/**99.9** |
| | ALL | 70.6/**11.9** | 31.8/**1.7** | 56.8/**98.0** | 88.7/**99.6** | 62.3/**98.4** | 90.7/**99.7** | 63.2/**97.4** | 80.2/**99.4** |
| CUSTOM NETWORK CIFAR-10 | ADVERSARIAL | 85.8/**36.7** | 85.3/**35.7** | 49.7/**87.7** | 75.2/**94.1** | 57.3/**87.9** | 79.9/**95.0** | 51.9/**85.7** | 69.2/**91.2** |
| | OOD | 77.7/**69.1** | 26.8/**5.4** | 70.6/**71.6** | 93.6/**98.6** | 72.5/**77.8** | 93.0/**98.6** | 69.1/**71.7** | 94.1/**98.7** |
| | ALL | 81.7/**52.9** | 56.1/**20.6** | 60.1/**79.7** | 84.4/**96.4** | 64.9/**82.9** | 86.5/**96.8** | 60.5/**78.7** | 81.6/**95.0** |
| WR-28-8 CIFAR-10 | ADVERSARIAL | 67.7/**33.5** | 68.7/**30.4** | 62.4/**91.2** | 79.1/**95.7** | 68.3/**91.9** | 81.9/**96.6** | 62.5/**89.3** | 67.3/**93.4** |
| | OOD | 64.8/**49.1** | 14.4/**1.2** | **76.0**/74.3 | 96.8/**99.6** | **77.3**/76.2 | 96.7/**99.7** | **76.6**/73.0 | 96.3/**99.6** |
| | ALL | 66.2/**41.3** | 41.5/**15.8** | 69.2/**82.7** | 87.9/**97.7** | 72.8/**84.0** | 89.3/**98.1** | 69.6/**81.2** | 81.8/**96.5** |

striking result is that for 3 out of the 4 settings, the logits KDE detector obtain better results than the softmax NN detector on adversarial examples. The only exception is on MNIST with the custom architecture, where the performances remains reasonably close (83.0% for KDE Logits, 86.6% for NN Softmax). It is quite remarkable that a logit-based detector with no prior knowledge on the illegitimate datasets performs overall better than a softmax-based one which has been exposed to these datasets during training.

Overall, these results show that *a significant part of the information that allows the detection of adversarial and OOD examples is definitely lost once the softmax function is applied*, as the results obtained with softmax probabilities are lower in almost all the considered settings, even with features tailored specifically for the detection task.

## 5 CONCLUSION AND PERSPECTIVES

We have shown through a series of experiments that the softmax function hinders the detection of adversarial and out-of-distribution, as it masks valuable information contained in the logits. We indeed observed that the normalization operated by the softmax forces the average value to be $1/C$ with $C$ the number of classes, although the average logit value is significantly different between legitimate and illegitimate examples. Other statistics than the average, such as the minimum and maximum, are also able to differentiate legitimate examples from the others when applied to logits, but we also saw that this separation is not present to the same extent with softmax probabilities, as using the exponential function pushes the highest value towards one and the lowest one towards 0.

We also showed quantitatively that this loss of information has a significant impact on detection performance. Indeed, detection methods that perform at levels competitive with the state of the art when using logit activations as inputs suffer from a considerable drop in performances when using softmax probabilities instead. This observation remains true on 2 datasets and 2 architectures, for a supervised condition where out-of-distribution and adversarial datasets are known in advance and for an unsupervised one.

These findings raise questions about the role of the softmax function in the lack of robustness observed in modern NN. We believe that further research on the impact of alternative activation functions can lead to interesting insights for this issue.

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

# A APPENDIX

## A.1 SCATTERPLOTS OF MINIMUM AND MAXIMUM LOGIT AND SOFTMAX VALUES

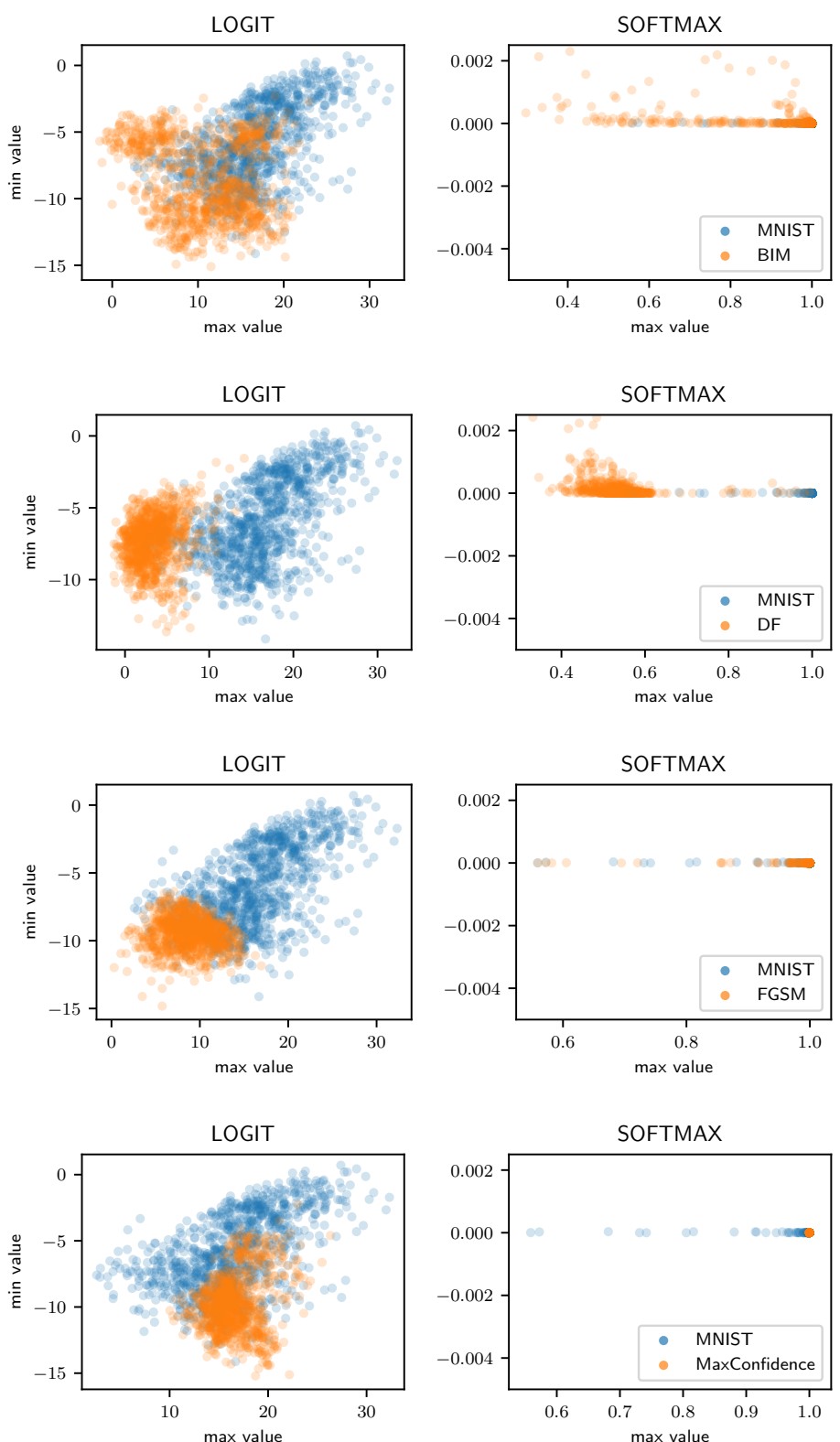

Figure 3: Scatterplot of the logit and softmax minimum and maximum values for adversarial examples. Best viewed in color.

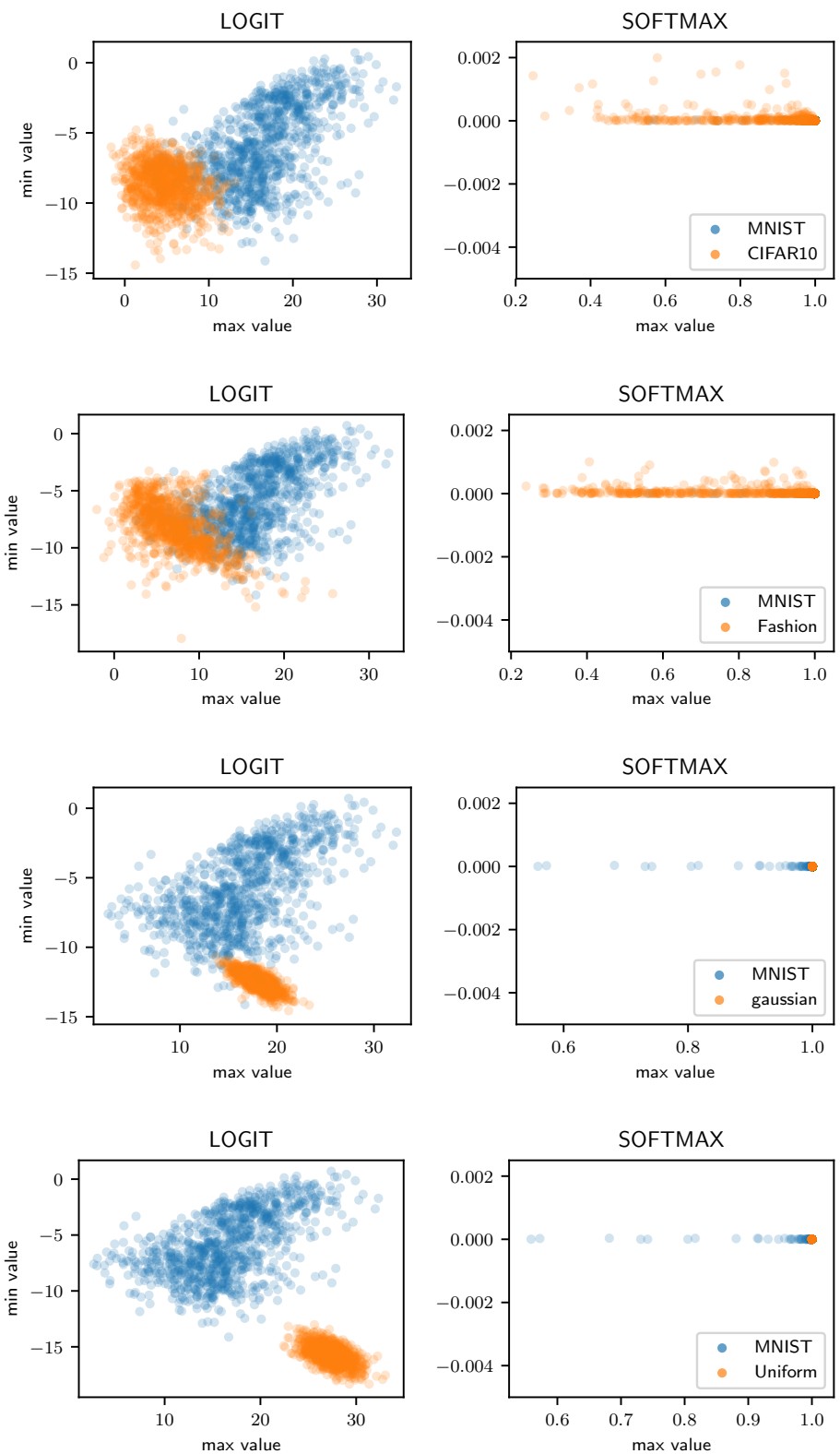

Figure 4: Scatterplot of the logit and softmax minimum and maximum values for OOD examples. Best viewed in color.

