# OpenReview forum: "How the Softmax Activation Hinders the Detection of Adversarial and Out-of-Distribution Examples in Neural Networks"
_ICLR.cc/2020/Conference — Reject_

### Official Review · AnonReviewer2 · 2019-10-21
**Official Blind Review #3**

**Rating:** 1

**Review:**

This paper suggests that the logits cary more information than the maximum softmax probability for OOD detection. They suggest this with scatterplots and develop techniques to support this claim.
They use logits for as features for OOD detection with by using a kernel density estimator. They also use a NN trained with logits features, but this assumes we can peak at the test distribution, so I ignore this entirely in this evaluation.
Unfortunately their KDE density estimator does not perform better than the maximum softmax probability for OOD detection on CIFAR-10 (74.3 vs 91.7).
They do not show results on CIFAR-100, but since the dimensionality of the logits would increase by an order of magnitude, one would expect kernel density estimation to perform much worse. The authors should include an evaluation on CIFAR-100 for completeness.

Small comments:

Table 3 is a comment about featurization. Does this hold when taking the log of the softmax probabilities (not the same as logits)? If not, then this isn't much a count against the softmax per se. Even then, this is a comment on using softmax information for KDE, not using the maximum softmax probability itself for OOD detection.

The full results for Table 2 are needed. Perhaps place this in an appendix.

They repeat that the logits contain more information than the maximum softmax probability, but so does the raw input. The challenge is not introducing more noise/variance when introducing more information.

I was confused at the experimental description. The information should be contained in one location. They train one of their CIFAR networks for 30 epochs, which isn't enough training time. Consequently, I suspect that those results are not worth drawing implications from since the accuracy is presumably low.


**Experience Assessment:**

I have published in this field for several years.

**Review Assessment: Checking Correctness Of Derivations And Theory:**

N/A

**Review Assessment: Checking Correctness Of Experiments:**

I carefully checked the experiments.

**Review Assessment: Thoroughness In Paper Reading:**

I read the paper at least twice and used my best judgement in assessing the paper.

---

> ### Author Response · Authors · 2019-11-12
> **Answer to Reviewer 3**
>
> Thank you for your comment and feedback. We respectfully disagree that the experiments with the NN-based detector should be ignored in your evaluation. As the goal of this paper is not to compare differences in performances between methods, but to compare differences in performances when using logit vs softmax values, we believe that the NN-based detector represents a best-case scenario where relevant features can be computed by the NN. We consider that it is interesting to observe that, even with an advantage as unfair as being able to peak at the test distribution, the softmax NN detector obtain overall worse results than the logits KDE detector on adversarial examples. Thus we argue that these experiments should be taken into account.
>
> We agree that it would be interesting to evaluate the KDE on datasets with more classes such as CIFAR-100. We will include this experiment in the next version of the paper.

---

### Official Review · AnonReviewer5 · 2019-10-30
**Official Blind Review #5**

**Rating:** 1

**Review:**

Summary

This paper showed that out-of-distribution and adversarial samples can be detected effectively if we utilize logits (without softmax activations). Based on this observation, the authors proposed 2-logit based detectors and showed that they outperform the detectors utilizing softmax activations using MNIST and CIFAR-10 datasets.

I’d like to recommend "reject" due to the following

The main observation (removing softmax activation can be useful for detecting abnormal samples) is a bit interesting (but not surprising) but there is no theoretical analysis for this. It would be better if the authors can provide the reason why softmax activation hinders the novelty detection.

The logit-based detectors proposed in the paper are simple variants of existing methods. Because of that, it is hard to say that technical contributions are very significant.

Questions

For evaluation, could the authors compare the performance with feature-based methods like Mahalanobis [1] and LID [2]?

I would be appreciated if the author can evaluate their hypothesis using various datasets like CIFAR-100, SVHN, and ImageNet.

[1] Lee, K., Lee, K., Lee, H. and Shin, J., 2018. A simple unified framework for detecting out-of-distribution samples and adversarial attacks. In Advances in Neural Information Processing Systems (pp. 7167-7177).

[2] Ma, X., Li, B., Wang, Y., Erfani, S.M., Wijewickrema, S., Schoenebeck, G., Song, D., Houle, M.E. and Bailey, J., 2018. Characterizing adversarial subspaces using local intrinsic dimensionality. arXiv preprint arXiv:1801.02613.

**Experience Assessment:**

I have published in this field for several years.

**Review Assessment: Checking Correctness Of Derivations And Theory:**

N/A

**Review Assessment: Checking Correctness Of Experiments:**

I carefully checked the experiments.

**Review Assessment: Thoroughness In Paper Reading:**

I read the paper thoroughly.

---

> ### Author Response · Authors · 2019-11-12
> **Answer to Reviewer 5**
>
> Thank you for you comments and feedback. We agree that our work would benefit from using more datasets for evaluation and comparing with more methods. We will include these improvements in the next version of the paper.

---

### Official Review · AnonReviewer4 · 2019-11-01
**Official Blind Review #4**

**Rating:** 3

**Review:**

This simple paper shows that the normalization of softmax causes a loss of information compared to using the unnormalized logits when trying to do OOD and adversarial example detection.  The main reason for this is of course the normalization used by the softmax. The paper is mostly empirical following this specific observation, and uses a number of examples on MNIST and CIFAR to show the improvement in performance by using unnormalized logits instead of softmax.

While interesting, it is to be noted that methods such as ODIN and temperature scaling specifically include a temperature to exactly overcome this same issue with softmax. The lack of comparison to such baselines makes this paper quite incomplete, especially as it is an empirical paper itself.

**Experience Assessment:**

I have published in this field for several years.

**Review Assessment: Checking Correctness Of Derivations And Theory:**

N/A

**Review Assessment: Checking Correctness Of Experiments:**

I carefully checked the experiments.

**Review Assessment: Thoroughness In Paper Reading:**

I read the paper thoroughly.

---

> ### Author Response · Authors · 2019-11-12
> **Answer to Reviewer 4**
>
> Thank you for your comments and feedback. Indeed, temperature scaling does limit the effect of using the exponential function to amplify differences in logits. However, it does not change the impact of the normalization, as the information about the logit absolute values is still lost after temperature. But we agree that this method and ODIN are interesting comparison baselines and we will include them in the next version of the paper.

---

### Decision · Program_Chairs · 2019-12-19

**Decision:**

Reject

**Comment:**


The paper investigates how the softmax activation hinders the detection of out-of-distribution examples.

All the reviewers felt that the paper requires more work before it can be accepted. In particular, the reviewers raised several concerns about theoretical justification, comparison to other existing methods, discussion of connection to existing methods and scalability to larger number of classes.

I encourage the authors to revise the draft based on the reviewers’ feedback and resubmit to a different venue.